# Exploration of the Different Dimensions of Wurtzite ZnO Structure Nanomaterials as Gas Sensors at Room Temperature

**DOI:** 10.3390/nano13202810

**Published:** 2023-10-23

**Authors:** Matshidiso P. Ramike, Patrick G. Ndungu, Messai A. Mamo

**Affiliations:** 1Department of Chemical Sciences, University of Johannesburg, Johannesburg 2028, South Africa; m.ramike@yahoo.com; 2Department of Chemistry, University of Pretoria, Private Bag X20, Hatfield, Pretoria 0028, South Africa; patrick.ndungu@up.ac.za

**Keywords:** semiconductor metal oxides, gas sensor, VOCs, polymer composites, sensor performance

## Abstract

In this work, we report on the synthesis of four morphologies of ZnO, namely, nanoparticles, nanorods, nanosheets, and nanoflowers, from a single precursor Zn(CH_3_COO)_2_·2H_2_O under different reaction conditions. The synthesised nanostructured materials were characterised using powder X-ray diffraction (XRD), Fourier transform infrared (FTIR) and Raman spectroscopy, UV–Vis, XPS analysis, transmission electron microscopy (TEM), scanning electron microscopy (SEM), and nitrogen sorption at 77 K. The XRD, FTIR, and Raman analyses did not reveal any significant differences among the nanostructures, but differences in the electronic properties were noted among the different morphologies. The TEM and SEM analyses confirmed the four different morphologies of the ZnO nanostructures. The textural characteristics revealed that the specific surface areas were different, being 1.3, 6.7, 12.7, and 26.8 m^2^/g for the nanoflowers, nanoparticles, nanorods, and nanosheets, respectively. The ZnO nanostructures were then mixed with carbon nanoparticles (CNPs) and cellulose acetate (CA) to make nanocomposites that were then used as sensing materials in solid-state sensors to detect methanol, ethanol, and isopropanol vapour at room temperature. The sensors’ responses were recorded in relative resistance. When detecting methanol, 6 out of 12 sensors were responsive, and the most sensitive sensor was the composite with a mass ratio of 1:1:1 of ZnO nanorods:CNPs:CA with a sensitivity of 0.7740 Ω ppm^−1^. Regarding the detection of ethanol vapour, 9 of the 12 sensors were responsive, and the 3:1:1 mass ratio with ZnO nanoparticles was the most sensitive at 4.3204 Ω ppm^−1^. Meanwhile, with isopropanol, 5 out of the 12 sensors were active and, with a sensitivity of 3.4539 Ω ppm^−1^, the ZnO nanoparticles in a 3:1:1 mass ratio were the most sensitive. Overall, the response of the sensors depended on the morphology of the nanostructured ZnO materials, the mass ratio of the sensing materials in the composites, and the type of analyte. The sensing mechanism was governed by the surface reaction on the sensing materials rather than pores hindering the analyte molecules from reaching the active site, since the pore size is larger than the kinetic diameter of the analyte molecules. Generally, the sensors responded well to the ethanol analyte, rather than methanol and isopropanol. This is due to ethanol molecules displaying a more enhanced electron-donating ability.

## 1. Introduction

Zinc-oxide (ZnO)-based gas sensors are among the most explored and commercially successful semiconducting metal oxide (SMOX) solid-state chemical sensors [1,2]. The key advantages of ZnO-based gas sensors are their chemical and thermal stability properties [3] and their ability to detect a variety of gases [2], such as methanol [4], ethanol [5], and isopropanol [6]. Methanol and ethanol are essential solvents in the chemical industry for manufacturing various industrial and household chemicals [7,8]. They quickly evaporate at room temperature and atmospheric pressure. As a result, individuals may be exposed to these organic liquids through inhalation, ingestion, and skin absorption. Short-term inhalation of these organic liquids can irritate the nose, throat, and lungs [7]. Long-term exposure to high concentrations can cause headaches, drowsiness, loss of consciousness, liver damage, and even death [7,8]. Therefore, the chemical industry requires a critical and continual need for fast, inexpensive, simple, and portable tools to detect these hazardous liquids.

The working principle of ZnO gas sensors is based on the variation in electrical conductance of the ZnO SMOX when exposed to various gases due to the reaction/s taking place between its surface layer and nearby gases [9]. However, at lower temperatures (≤100 °C), the variation in conductivity of the ZnO SMOX is insignificant because the surface reactions occur very slowly [10,11]. Therefore, traditional ZnO gas sensors operate at high temperatures within the 300–500 °C range [12,13,14] to provide the necessary activation energy required for the redox surface reactions and increase the kinetics of the response mechanism, thus making sensing measurements possible [13,14].

Conversely, the high operating temperature is the most critical flaw of ZnO SMOX gas sensors. It leads to increased power consumption and the degradation of materials over time and can be a safety hazard when detecting a target analyte in a flammable and explosive gas environment [12,13,14]. Some effective strategies to overcome this flaw include modifying the morphology (form, shape, or structure) of the ZnO nanostructure and reducing the grain and/or particle size to values that are comparable to the Debye length [15,16,17,18]. The incorporation of noble metals [19], polymers [12], metal oxides [20], or conducting carbon matrices [3,13] into the pure ZnO nanostructures [14,17] has led to sensors that can work at lower temperatures.

ZnO nanostructures have a large surface-to-volume ratio, numerous surface active sites, and a high electron mobility compared to other metal oxides [14,17,21]. Therefore, redox surface reactions are accelerated, and the operating temperature is reduced [14]. The physical, chemical, electronic, optical, and catalytic properties of the ZnO nanostructures depend on their morphology, size, and specific surface area. Therefore, modifying the morphologies of the ZnO nanostructures affects not only their operating temperature but also their gas-sensing performance [22].

Most studies have used conducting polymers to overcome the high temperature of the ZnO-based gas sensor by incorporating a polymer into the ZnO structure [14,22]. Sensors based on conducting polymers display a high sensitivity and short response time and, in most cases, operate at room temperature [14]. However, cellulose acetate, an insulating, bio-based polymer, was used in this study. Cellulose acetate has several advantages, including non-toxicity and its resistance to alkali, acid, thermal, and chemical corrosion [23].

Candle soot is the carbon-rich black residue from burning a candle [24,25]. Several studies have demonstrated candle soot as a source of carbon nanoparticles with different optical, physical, and chemical properties [26,27,28]. These carbon nanoparticles find use in various applications, including sensing, due to their many appealing properties, such as their nano-sized dimensions, high surface area, and electronic conductivity [24].

This study synthesised ZnO nanostructures with different morphologies (i.e., particles, rods, sheets, and flowers) and incorporated them with cellulose acetate and carbon nanoparticles generated from candle soot. The resultant composite material was used as the sensing material on interdigitated gold-coated electrodes towards methanol, ethanol, and isopropanol sensors without an additional heat source and run at room temperature and atmospheric pressure.

## 2. Experimental: Materials and Method

### 2.1. Synthesis of Nanomaterials

In this investigation, the chemicals were purchased in South Africa from Associated Chemical Enterprises (ACE) and used as received without further purification. The white candles (Lighthouse Candles) were purchased from a local supermarket in Johannesburg, South Africa.

#### 2.1.1. Synthesis of Carbon Nanoparticles

The carbon nanoparticles were synthesised following a simple and inexpensive method comprising the incomplete combustion of white paraffin wax candles [24]. Briefly, a ceramic mug was positioned about a centimetre above the burning white candles to collect the soot released from the flame. When sufficient soot was collected, the product was collected and then purified via centrifugation using absolute ethanol, followed by drying at 60 °C in an oven for 12 h.

#### 2.1.2. Synthesis of Zinc Oxide Nanoparticles (Zero-Dimensional)

The zinc oxide nanoparticles were prepared using a direct precipitation method [29]. Typically, aqueous solutions of 0.04 mol Zn(CH_3_COO)_2_·2H_2_O and 0.08 mol NaOH were prepared separately at room temperature. The NaOH solution was added dropwise into the zinc acetate solution under vigorous stirring at room temperature. The resultant white suspension was separated and washed several times with deionised water using a centrifuge and subsequently dried at 60 °C overnight before calcining at 500 °C in an air atmosphere using a muffle furnace.

#### 2.1.3. Synthesis of Zinc Oxide Nanorods (One-Dimensional)

The zinc oxide nanorods were synthesised via the thermal decomposition of zinc acetate dihydrate [30]. In the procedure, 0.5 g of Zn(CH_3_COO)_2_·2H_2_O was added into an alumina crucible, covered with a lid, and heated to 300 °C in a muffle furnace for 12 h in air. After 12 h, the sample was left to cool to room temperature naturally. The resultant ZnO nanorods in powder form were used as they were, without purification.

#### 2.1.4. Synthesis of Zinc Oxide Nanosheets (Two-Dimensional)

The zinc oxide nanosheets were synthesised via the room-temperature precipitation procedure described by Li et al. [31]. To 100 mL of an aqueous solution of 0.015 mol Zn(CH_3_COO)_2_·2H_2_O, 75 mL of 0.08 mol NaOH solution was added dropwise under continuous moderate magnetic stirring. Medium stirring was maintained for 30 min after the addition of the NaOH aqueous solution. After 30 min, stirring was stopped, and the milky solution was left undisturbed for 30 min at room temperature. The product was filtered and washed alternatively with deionised water and absolute ethanol several times using a centrifuge. The resultant white ZnO nanosheet powder was then dried at 60 °C overnight and calcined at 200 °C in an air atmosphere for 30 min.

#### 2.1.5. Synthesis of Zinc Oxide Nanoflowers

The zinc oxide nanoflowers were synthesised via a hydrothermal reaction described by Lu et al. [32]. In a typical synthesis method, 7.3 mmol of Zn(CH_3_COO)_2_·2H_2_O were dissolved in 10 mL of absolute ethanol and 20 mL of deionised water. The pH of the solution was adjusted to 10 under vigorous stirring using a 25% ammonia solution. The subsequent milky white slurry was transferred into a 50 mL Teflon-lined autoclave and heated at 140 °C for 10 h in an oven. Then, the solution was allowed to cool down to room temperature, and then the synthesised product was collected and washed with deionised water and then absolute ethanol, several times each, using a centrifuge. Subsequently, the resultant white product was dried at 60 °C for four hours before calcining at 500 °C in a furnace for 2 h.

### 2.2. Composite Preparation for the Different ZnO-Based Gas Sensors

Twelve sensors were fabricated via the weighing of 30 mg, 20 mg, and 10 mg of each of the four synthesised ZnO nanostructures into different small sample vials. Then, into each sample vial, 10 mg of the candle soot and 10 mg of cellulose acetate were mixed, and then 5 mL of N, N dimethyl formamide (DMF) was added to the mixture as a solvent. The resultant mixtures were dispersed for 30 min via sonication and then moderately stirred for 48 h using a magnetic stirrer plate. Subsequently, 5 µL of each resulting homogeneous mixture was coated evenly on an interdigitated gold-coated electrode using a micropipette. The fabricated sensors were air-dried in a dust-free environment at room temperature and then maintained in a vacuum desiccator for seven days before electrical characterisation. The 3:1:1, 2:1:1, and 1:1:1 mass ratios of nanocomposite sensors were prepared for all the ZnO morphologies used.

### 2.3. Characterisation Techniques

The synthesised ZnO nanostructure morphology was confirmed using a JEM-2100 transmission electron microscope (TEM) and Vega 3 Tescan scanning electron microscope (SEM). The TEM and SEM electron microscopes also provided information about the structure, size, and complexity of the surfaces of all the nanomaterials in this investigation. The identity and purity information was acquired using a Phillips X-ray diffractometer from PANalytical (X’ Pert Pro core) using a 2ϴ scanning range of 5–90° with a Cu Kα (λ = 0.154 nm) radiation source. Chemical bonding within the nanomaterial was further explored using a Raman microscope (WITec alpha 300R confocal fluorescence microscopy, 532 nm wavelength) and an FTIR spectrometer from PerkinElmer (TL 8000 Balanced flow FT-IR EGA system spectrometer), and the obtained data were baseline-corrected and normalised. The synthesised materials’ band gap energy (e.g., eV) was determined using their diffuse reflectance spectra measured with a UV–Vis Spectrophotometer from Thermo Scientific (EvolutionPro). A Micromeritics ASAP 2460 surface area and porosity analyser was used to determine the surface area and porosity of the nanomaterial. X-ray photoelectron spectroscopy (XPS, XSAM800, Kratos, Manchester, UK) was used to determine the oxidation state and the elemental analysis.

### 2.4. The System for the Electrical Characterisation of the Sensors

A 20 L four-necked round-bottom flask was used as the reaction chamber for the prepared sensors to interact, alternatively, with the analyte and air atmosphere. The first neck of the 20 L round bottom flask was used to suspend the sensor inside the reaction chamber. The sensor was connected with two insulated wires to the LCR meter (Keysight E4980AL LCR Meter 20 Hz–300 kHz). The sensor responses were measured at 0.5V AC and 25 kHz [33]. The second neck was used as a fresh air inlet into the reaction chamber during the recovery phase. The third neck was used to remove the gas analyte after exposure to the sensor using an oilless piston vacuum pump at atmospheric pressure. The fourth neck of the flask was used to introduce the volatile analytes as a liquid using a micro-syringe (see Appendix A).

The sensors were run under air (without analyte gas) for 5 min, and then the volatile analyte’s first volume (e.g., 5 µL) was injected into the reaction chamber. The sensor was exposed to the evaporated analyte for 10 min, and then the remaining analyte was flushed out using a vacuum pump for 3 min under atmospheric pressure. Thus, the sensor could recover before the subsequent incremental volumes were injected, and this was repeated until all the intended analyte vapour exposures were completed. Each sensor was exposed to one analyte at a time in the increasing volumes of 5, 10, 15, 20, 25, and 30 µL.

The concentrations of the volatile analytes were calculated in ppm using the following equation [34,35] from the injected volume:$$C=\frac{22.4\;\rho TV_{s}}{273\;MV}\times 1000$$
where C is the concentration of the exposed evaporated analyte in (ppm), ρ is the density of the liquid form of the volatile analyte in (g mL^−1^), T is the temperature of the environment during analysis in (K), Vs is the volume of the injected liquid form of the volatile analyte in (μL), M is the molecular weight of the volatile analyte in (g·mol^−1^), and V is the volume of the reaction chamber in (L).

The electrical responses of the prepared sensors to the target gas were reported as the relative resistance (ΔR) and defined as
ΔR = R_gas_ − R_air_
where R_gas_ and R_air_ are the resistance of the sensors when exposed to the target gas and air atmosphere, respectively; all the sensors’ performance measurements were conducted at room temperature, atmospheric pressure, and a relative humidity of around 42%.

### 2.5. Sensitivity, Response, and Recovery Time of the Sensors

The sensitivity of the fabricated sensors was measured from the slope of the linear fitted curve of each sensor; it was calculated from the derivative of the sensor response (ΔR) against the change in analyte vapour concentration. The response time measures the time required for the sensor to generate an equivalent signal to the specific analyte vapour concentration to which it is exposed. The response time is considered as the time needed to reach 90% of the maximum signal, and the recovery time is the time required to recover from 90% of the maximum signal [36].

## 3. Results and Discussion

### 3.1. XRD Analysis

The synthesised nanomaterial and purchased cellulose acetate were analysed with the X-ray powder diffractometer to determine their identity, purity, and crystallinity. The resultant patterns are shown in Figure 1. The XRD patterns of the four synthesised ZnO nanostructures exhibited similar distinctive sharp peaks at equivalent angles. The diffraction peaks were located at 2θ = 31.5°, 34.2°, 36.0°, 47.4°, 56.4°, 62.5°, 66.2°, 67.8°, 68.8°, 72.9°, and 76.6°, which correspond to the lattice planes of (100), (002), (101), (102), (110), (103), (200), (112), (201), (004), and (202), respectively. All the diffraction peaks are indexed to the hexagonal wurtzite ZnO standard, consistent with JCPDS card no. 36-1451 [37,38]. Nanomaterials prepared with different morphologies should show some changes in the XRD pattern, mainly due to the preferred orientation of the nanoscale materials as they are prepared for XRD analysis [39]. The XRD patterns of the ZnO sample prepared in this study do show that the relative intensities of the (100), (002), and (101) have some differences among the NF, NS, NR, and NP ZnO samples, which indicates that the different morphologies favour different crystal planes. This observation is an initial indication that the samples have different morphologies.

The average crystallite sizes of the synthesised ZnO nanoparticles, nanorods, nanosheets, and nanoflowers were calculated using the Scherrer equation:D = 0.9λ/(β cosθ)
where D is the crystal size (nm), λ is the wavelength of the X-rays, i.e., Cu Kα (0.154 nm), β is the full width at half the maximum height of the diffraction peak (radians), and θ is the Bragg diffraction angle of the corresponding peak in degrees [25]. The crystallite sizes of the as-prepared ZnO nanoparticles, nanorods, nanosheets, and nanoflowers were 35 nm, 41 nm, 28 nm, and 44 nm, respectively, when calculated using the (1 0 0) reflection peak of each XRD pattern.

Candle soot exhibited two diffraction peaks: a distinctive peak at 2θ = 25.0°, corresponding to the (002) crystal planes associated with the diffraction of the repeated plane of graphite [25]; and the relatively low-intensity (111) reflection peak at 2θ = 43.9°, which is attributed to the diamond phase in the candle soot [25]. The XRD patterns of the purchased cellulose acetate (Figure 2c) demonstrated the characteristic peaks for cellulose acetate at 2θ = 8.8°, 10.5°, 13.5°, 17,4°, 21,5 °, 35,1°, and 44,8° [40,41].

### 3.2. FTIR Spectroscopy Characterisation

Appendix A illustrates the FTIR spectra of the synthesised ZnO nanostructures measured in the 4000–400 cm^−1^ spectral range. The FTIR spectra of the four as-synthesised ZnO nanostructures showed two strong bands of varying intensities at ~510 cm^−1^ and 420 cm^−1^, attributed to the Zn–O stretching bonds [42]. A broad band at ~3400 cm^−1^ was attributed to the O-H stretching vibrations and bending modes on the surface of ZnO [43,44]. The ZnO wurtzite crystal structure has two optical phonon modes that are IR active, the A1 and E1 modes; the slight shifts in the two peaks in the samples can be attributed to surface phonon modes due to the nano-size and morphology of the different samples. Similar shifts and changes regarding the FTIR peaks for ZnO have been reported in the literature [45,46,47,48]. The IR bands at ~1400 cm^−1^ and ~1630 cm^−1^ were attributed to the symmetric and asymmetric stretching modes of the carbonyl groups (-COOH) from organic reaction intermediates such as (Zn_4_O(CH_3_COO)_6_) [30,43,44]. The FTIR spectrum of the generated candle soot exhibited three prominent absorption bands at 3459 cm^−1^, 1630 cm^−1^, and 611 cm^−1^ (Appendix A). The band at 3459 cm^−1^ was ascribed to O-H bond stretching vibrations due to the water adsorbed [24,25,26]. The 1630 cm^−1^ and 611 cm^−1^ bands were attributed to the C=C bond stretching vibration [26] and sp2 and sp3 aromatic carbon soot, respectively. The FTIR spectrum of the purchased cellulose acetate (Appendix A) had four oxygen bands at 3470 cm^−1^, 1635 cm^−1^, 1250 cm^−1^, and 1039 cm^−1^, corresponding to the -OH stretching of the unacetylated cellulose, the H-O-H bending of absorbed water, the C-O stretching of the acetyl group C-O, and the C-O-C bond of the cellulose backbone, respectively [39]. In addition, 1392 cm^−1^ and 625 cm^−1^ bands correspond to the C-H bending vibration of CH_3_ in the acetyl group.

### 3.3. Raman Spectroscopy Characterisation

ZnO has a wurtzite crystal structure in the $C_{6v\;}^{4}$(P6_3_mc) space group. The Brillouin zone of the phonon modes at Γ point are given by Γ_opt_ = 1A_1_ + 2B_1_ + 1E_1_ + 2E_2_ where both A_1_ and E_1_ modes are polar. They are further grouped into transverse optical (A_1_ TO and E_1_ TO) and longitudinal optical (A_1_ L.O. and E_1_ L.O.) modes. Furthermore, the E_2_ mode has a low (E_2_ ^low^)- and a high (E_2_ ^high^)-frequency phonon mode, due to the vibration of oxygen atoms and heavy Zn sublattices, respectively [44,45,46,49,50]. However, both the B_1_ branches B_1_ (low) and B_1_ (high) modes are Raman-inactive. Figure 2a–e show the Raman spectra of the ZnO nanoflowers, nanosheets, nanorods, and nanoparticles, and carbon nanoparticles. For the ZnO different morphology, peaks for $E_{2}^{low}$ and $E_{2}^{high}$ usually appear around 100 cm^−1^ and 443 cm^−1^. For the $E_{2}^{low}$ phonon mode, blue shifts of 1 cm^−1^ and 2 cm^−1^ were observed for the nanoflower nanostructure for nanorods, and red shifts of 1 cm^−1^ were observed for nanosheets and nanoparticles. However, the main dominant $E_{2}^{high}$ peak at 437 cm^−1^ was observed at higher frequencies for all morphologies, with a blue shift of 6 cm^−1^, 5 cm^−1^, 4 cm^−1^ and 5 cm^−1^ for the nanoflowers, nanosheet, nanorods and nanoparticles, respectively. There could be many reasons why the blue shift in $E_{2}^{high}$ was observed compared to the theoretical values (see Table 1). One of the reasons could be the optical phonon confinement by nanostructures, as the nanostructures are bigger in size than the Bohr exciton radius in ZnO. The second reason that could contribute to the blue shift is due to different growth directions leading to anisotropic internal strains, resulting in phonon localisation by defects or impurities in the nanostructures [44,45,46,49,50]. The Raman frequency of the E_2_ ^high^–E_2_ ^low^ phonon blue-shifted by at least 2 cm^−1^ from the theoretical value for the nanoflowers. A peak at around 585 cm^−1^ is due to ZnO’s 1E1 (L.O.) mode, which is associated with oxygen deficiency [51]. The Raman frequency of the 1E_1_(L.O.) is blue-shifted from the theoretical values. The peak position at 208 cm^−1^ is due to the second-order *2*$E_{2}^{low}$ [52], and the peaks at around 384 cm^−1^, 390 cm^−1^, 389 cm^−1^, and 387 cm^−1^ are attributed to the A_1_(TO) mode [53] for the nanoflowers, nanosheets, nanorods, and nanoparticles. However, all the observed A_1_(TO) modes are blue-shifted by at least 4 cm^−1^ from the theoretical value. A less-intense broad peak at 683 cm^−1^ is due to the unreacted acetate groups and is assigned to the O-C-O symmetric bends [54].

As shown in Figure 2e, the Raman spectrum of carbon nanoparticles has two prominent peaks assigned to the first-order D mode at 1338 cm^−1^ and G mode at 1587 cm^−1^. The D mode, A1g symmetry, is the in-plane breathing vibration of aromatic rings moving away from the Brillouin zone centre. In a perfect graphitic lattice, this band is forbidden; however, it becomes active when the materials are exposed to chemical reactions on the surface, leading to disorder. However, the G mode E2g symmetry is due to the in-plane stretching vibration of the carbon atom pairs [24,25,26,45,46].

### 3.4. XPS Analysis

The XPS survey scans of the ZnO NRs, ZnO NFs, ZnO NSs, ZnO NPs, and CNPs are shown in Figure 3. The photoelectron peaks of the main elements, Zn, O, and C, were obtained. The XPS studies were performed to determine Zn’s chemical states in ZnO composites. The two strong peaks centred at around 1022 eV and 1045 eV were observed for all four morphologies (see Figure 4), which are in agreement with the binding energies of Zn 2p^3/2^ and Zn 2p^1/2^, respectively, and this confirms the Zn ion in the +2 oxidation state as Zn^2+^ [55]. Three Gaussian components fitted the broad O 1s peaks for all samples at around 530 eV (O_α_), 531 eV (O_β_), and 533 eV (Oγ), respectively, as shown in Figure 5 and Table 2 The three oxygen species were present for all the samples except for CNPs; only the O_α_ and O_β_ were observed. The O_α_ component, with ZnO nanostructures, has been identified in the literature and is due to the O^2−^ ions in the hexagonal wurtzite structure, which are surrounded by a Zn^2+^ ion array [55]. The available oxygen species of the Oγ and Oβ increased significantly in the composites, except the ZnO NR-CNP-CA, which dropped by 50% of the Oγ from 11% to 6% (see Table 3). Both the Oγ and Oβ are critical oxygen species in sensing.

### 3.5. TEM Characterisation

Figure 6a shows the TEM images of the as-synthesised ZnO nanoparticles. The ZnO nanoparticles were of varying shapes and sizes, mainly spherical, with an average diameter of 33.03 nm, which is close to the 35 nm crystallite size determined from the XRD measurements. The TEM image of the ZnO nanorods confirmed their rod-like morphology (Figure 6b). The average diameter of the as-synthesised ZnO nanorods was 36.11 nm, comparable to the 41 nm crystallite size calculated from the XRD patterns. The as-synthesized ZnO nanosheets were thin and perforated, as demonstrated by their TEM image (Figure 6c). The perforation of the nanosheets may increase the specific surface area [31]. Figure 5d shows the rods assembled in a flower-like structure, thus verifying the morphology of the as-synthesised ZnO nanorod flowers. The TEM image of the candle soot (Figure 6e) exhibited clusters of round-shaped nanoparticles. The purchased cellulose acetate had a sheet-like morphology (Figure 6f).

### 3.6. FE SEM Characterisation

Figure 7 shows the FE SEM images of the as-synthesised ZnO nanostructures and their different morphologies. The ZnO nanoparticles displayed different shapes, including spherical, oval, and irregular shapes. However, the ZnO nanorods have similar shapes and lengths, similar rod-like structures, and similar sizes. The ZnO NSs have disc-like structures with similar dimensions, about 40 nm in thickness. The ZnO NFs are displayed in Figure 7d as star-like 3D flower structures with a distinct uniform morphology ranging from 3 to 7 µm.

### 3.7. Textural Characteristics of the Synthesised ZnO Nanostructure Samples

The surface area, porous volume, and diameters of the synthesised nanomaterials were determined from the nitrogen adsorption and desorption isotherm (BET). The surface area of the nanoparticles, nanorods, nanosheets, and nanoflowers was 6.7 m^2^/g, 12.70 m^2^/g, 26.8 m^2^/g, and 1.3 m^2^/g, respectively (see Table 3). As expected, the perforated ZnO nanosheets had the largest surface area of the four ZnO nanostructures. The specific surface area of the candle soot and purchased cellulose acetate was 84.82 m^2^/g and 2.63 m^2^/g, respectively. The specific surface area of the various nanocomposites was between 30.1 and 36.7 m^2^/g; the highest was for the ZnO NS composite, and the lowest was for the ZnO NF composite. Regarding the pore volume and pore size, ZnO NPs had the largest pore volume and pore size; the lowest pore volume was for ZnO NFs, and the smallest pore size was recorded for ZnO NSs.

### 3.8. The Gas-Sensing Properties

The composites were prepared via varying of the mass of metal oxide in the composite while the mass of the CNP and of the polymer were kept constant. A total of twelve sensors were prepared and then exposed to three organic analytes, namely, methanol, ethanol, and isopropanol vapours, at room temperature and atmospheric pressure. The performance of the fabricated sensors was evaluated using their sensitivity, response, and recovery time.

All the active sensors based on the three sensing materials showed a decreased resistance during exposure to the analyte vapour and an increased resistance when the analyte vapour was removed from the system and returned to the baseline. This indicates that the fabricated sensors can regenerate themselves without the analyte vapour. All the active sensors showed a linear relationship between their relative responses and the analyte vapour concentrations. The non-active (non-responsive) sensors are those that display a very high noise-to-signal ratio.

Not all the prepared sensors responded to the methanol vapour; only six responded out of the 12 prepared sensors. The sensors with the mass ratios of 1:1:1, 2:1:1, and 3:1:1 of the ZnO-nanorod-based sensors, the 1:1:1 ZnO nanosheet, and the 1:1:1 and 3:1:1 ZnO nanoflower composites were active. All the sensors based on ZnO nanoparticles generally did not respond to the methanol vapour. The relative resistance of the sensors linearly decreased as the concentration of the methanol vapour increased (see Figure 8) with negative gradients. The response curves exhibited saturation plateaus in most cases, and the sensors were noted to display p-type semiconductor behaviour.

The sensitivity of the sensors towards methanol vapour was strongly related to the mass ratio of the metal oxide. For example, for nanorod ZnO, the highest sensitivity was for the lowest mass ratio, 1:1:1, of the composite’s sensing materials, followed by the 2:1:1 mass ratio, and the lowest sensitivity was for the 3:1:1 mass ratio (see Table 4). Like the nanorod-ZnO-based sensors, the nanoflower-structured-ZnO-based sensor recorded a higher sensitivity in the lowest mass ratio. In the case of the nanosheet-based sensors, only the sensor based on a 1:1:1 mass ratio responded to methanol vapour.

Generally, sensors with a low mass ratio responded well and had better sensitivity. Of all the sensors fabricated, the nanorod-structured ZnO sensor with a 1:1:1 mass ratio recorded the highest sensitivity to methanol, at 0.7740 Ω ppm^−1^. The lowest sensitivity was recorded for the nanorod-structured ZnO sensor at a 3:1:1 mass ratio, at 0.06437 Ω ppm^−1^, which is over ten times less than the sensitivity of the sensor with a 1:1:1 mass ratio.

The same fabricated sensors were exposed to ethanol vapour, and the sensors’ performance was investigated. Out of the twelve fabricated sensors, only nine sensors responded to the ethanol vapour. The sensor based on 3:1:1 mass ratio ZnO nanoparticles responded and exhibited the highest sensitivity (4.3204 Ω ppm^−1^) towards the ethanol vapour of all nine fabricated sensors (see Appendix A).

All the nanorod-structured-ZnO-based sensors did respond towards ethanol vapour, and the sensitivity of the ZnO-nanorod-composite-based sensors towards ethanol vapour showed an inverse relationship between the amount of ZnO in the composite and the sensitivity of the sensor. At a low mass ratio in the composite, i.e., a 1:1:1 mass ratio, the sensor sensitivity was higher than the 2:1:1 and 3:1:1 ZnO nanorod composite sensors (see Appendix A). In the case of nanosheet-structured-ZnO-based sensors, except for the sensor based on a 3:1:1 mass ratio, the two prepared sensors responded well to the ethanol vapour. In terms of sensitivity, the sensor based on a 1:1:1 mass ratio had a better sensitivity (1.1067 Ω ppm^−1^) than the sensor based on a 2:1:1 mass ratio, which had a sensitivity of 0.3362 Ω ppm^−1^ (Appendix A).

For the nanoflower-structure-based sensors, all the sensors with different mass ratios responded well to ethanol vapour. A high sensitivity was observed with the ZnO structured as nanoflowers with a 1:1:1 mass ratio (1.9670 Ω ppm^−1^), while the 3:1:1 mass ratio sensor had a sensitivity of 0.8520 Ω ppm^−1^ and, finally, the 2:1:1 mass ratio sensor had the lowest sensitivity (almost a quarter of that of the 1:1:1 mass ratio sensor), with a value of 0.5643 Ω ppm^−1^.

In the case of isopropanol, only five sensors out of the twelve produced a response indicative of a sensor. With the nanoparticle-structured-ZnO-based sensors, only that based on a 3:1:1 mass ratio was responsive, with a sensitivity of 3.4539 Ω ppm^−1^. The other sensors, with 2:1:1 and 1:1:1 mass ratios, did not respond (see Appendix A). For the nanorod-structured-ZnO-based sensors, only the 2:1:1 mass ratio did respond, with 0.8077 Ω ppm^−1^ sensitivity. For the nanosheet-structured-ZnO-based sensors, only those with 2:1:1 and 1:1:1 mass ratios responded, with sensitivities of 0.5070 and 0.8780 Ω ppm^−1^. Finally, in the nanoflower-structured ZnO sensors, only the 1:1:1 mass ratio responded, with a sensitivity of 2.4753 Ω ppm^−1^. The highest sensitivity towards the isopropanol vapour was that of the sensor based on 3:1:1 mass ratio nanoparticles, followed by the nanoflower-structured-ZnO-based sensor based on a 1:1:1 mass ratio, and the lowest sensitivity observed was in the nanoflower-based sensor with a 2:1:1 mass ratio. Generally, all the fabricated sensors performed poorly towards the isopropanol vapour.

The stability of the sensors was examined over five cycles. The test was conducted by exposing the sensor to 300.3 ppm of methanol analyte and, once the response reached a stable plateau, fresh air was injected to remove the analyte from the chamber. This process was repeated five times. The sensor response was 750 ± 11 Ω over the five cycles, which indicates the sensor is very stable (see Appendix A).

The sensing materials usually have porous structures, through which the analyte gas passes and reacts with the adsorbed reactive oxygen species on the surface of the materials. The concentration of the analyte gas will decrease faster with a longer diffusion path than a shorter one. The diffusion and surface reaction rate usually determine the analyte gas concentration in the porous structure. The diffusion rate and mechanism of the analyte gas through the porous sensing material largely depend on the pore size and surface diffusion. There are generally four types of diffusion mechanisms through porous materials; the first one is bulk Poiseuille flow, which is more dominant when the pore sizes are larger than the mean free path [56]; the second type is Knudsen diffusion, in which the gas molecules pass through long and narrow pores with a diameter between 2 and 50 nm [57]. The third type is surface diffusion, which becomes more dominant when the analyte gas molecules are in a strong potential field relative to the pore walls. The molecules then tend to be adsorbed on the walls of the porous structure and thus move out of the gaseous phase. Therefore, the adsorbed molecules vibrate at the adsorption site and diffuse slowly to the next location [58]. The final type is gas translational diffusion, which often occurs in small pore-size materials. In gas translational diffusion, when the analyte gas molecules gain enough kinetic energy to escape the surface potential field of one side of the pore wall, the other side stops them. Usually, the molecules diffuse through by jumping from one site to a second site [59]. Our synthesised materials have pore sizes ranging from 204 to 196 nm (see Table 3), and the kinetic diameter of the analyte molecule is 0.36 nm, 0.43 nm, and 0.47 nm for methanol, ethanol, and isopropanol, respectively [60,61]. The pore size of the synthesised materials is much larger than the kinetic diameter of the analyte molecule, and there is expected to be no hindrance during diffusion through the pores. Therefore, the diffusion mechanism is a bulk Poiseuille flow for the sensing materials. According to our results, the sensors that are more sensitive to ethanol and less sensitive to methanol and isopropanol are not expected to be dominated by pores functioning via a sieving-type mechanism but, rather, the dominant mechanism is likely due to reactivity on the surface of the sensing materials.

This phenomenon has been explained using the reduction–oxidation sensing mechanism; ethanol displays a more enhanced electron-donating ability than methanol [62]. Therefore, when ethanol interacts with the surface of a typical n-type semiconductor, such as ZnO, the measured changes in conductivity and sensitivity are enhanced [63].

Overall, the ZnO-nanorod-based sensors’ performances surpassed those of the other fabricated sensors (see Figure 9). One-dimensional ZnO nanostructures have been reported to exhibit excellent conductivity [64].

Regarding the response and recovery time for the active (responsive) sensors, the response and the recovery time during exposure to methanol vapour was between 2 and 3 min, except for the nanosheet- and nanoflower-based sensors, for which it was over 3 min (see Table 4). Interestingly, all the mass ratios of the nanorod-based sensors did respond well, and we noticed that the sensors with high mass ratios tended to respond fast, while the sensors with low mass ratios responded slowly. Such a trend was observed for the recovery time, as well. A similar trend was observed during the exposure of the sensors to ethanol vapour; the response and recovery times were between 2 and 3 min, with some exceptions: for a mass ratio of 2:1:1 nanoflower, the response time was about 1.5 min, and it was 6.4 min for a mass ratio of 3:1:1 nanorod. The fastest recovery was recorded for the nanorod with a 3:1:1 mass ratio at 1.2 min, and the slowest recovery was recorded for the nanosheet with a mass ratio 2:1:1 at 3.5 min. The relationship we observed between the mass ratio and the response time for nanorod-structure-based sensors towards ethanol vapour was that when the mass ratio increased in the composite, the response time became slower; this effect was the opposite for methanol vapour.

### 3.9. The Gas-Sensing Mechanism

A ZnO-based gas sensor’s gas-sensing behaviour depends on the redox reactions between the chemically adsorbed oxygen species and analyte vapour on the zinc oxide surface [31]. When the ZnO-based gas sensor is exposed to atmospheric air, oxygen is adsorbed on the ZnO surface layer [10,11,31]. The adsorbed oxygen interacts with the ZnO surface, removing electrons from the conduction band of the ZnO material and forming charged oxygen species such as O^2−^, O^−^, and O^2−^ [11,36,65], thus creating an electron-depleted layer [36,66]. Once the electron-depletion layer is formed, the flow of the electrical current of an n-type semiconductor, such as ZnO, decreases as the measured resistance increases. When the ZnO-based gas sensor is then exposed to reducing gases, such as methanol, ethanol, and isopropanol, the analyte gas reacts with the charged oxygen species and releases electrons back into the conduction band of the ZnO material. As a result, the flow of the electric current increases, and the measured resistance decreases. The measured resistance of all the fabricated sensors decreased when exposed to the methanol, ethanol, and isopropanol vapours. This behaviour demonstrates that the synthesised sensing materials have an n-type conductivity.

## 4. Conclusions

All four morphologies of ZnO—nanoparticles, nanorods, nanosheets, and nanoflowers—were synthesised from the single precursor Zn(CH_3_COO)_2_·2H_2_O under different reaction conditions. Indexing the diffraction patterns of the ZnO nanostructures confirmed that all materials had a hexagonal wurtzite structure, and differences in the relative intensities of (100), (002), and (101) provided some initial indication that the different morphologies had been achieved. The morphologies of the synthesised nanostructured materials were confirmed via TEM and SEM analysis, with the ZnO nanoparticles showing a range of spherical and irregular shapes; the ZnO nanorods displayed rod-like morphologies, whereas the ZnO nanosheets displayed a 2D sheet-like morphology, and the nanoflowers had a star-like 3D structure. FTIR analysis did not reveal any significant differences among the various ZnO nanostructures, whereas the Raman results did show minor peak shifts from the bulk ZnO values reported in the literature. The XPS results revealed an increase in the key oxygen species involved in the sensing mechanism; specifically, the O_β_ and O_γ_ relative percentages increased from those observed with individual ZnO nanostructures relative to the values observed with the nanocomposites.

The synthesised materials with carbon nanoparticles and CA as composite with variable mass ratios of ZnO nanostructures were used as sensing materials in solid-state sensors to detect methanol, ethanol, and isopropanol vapour at room temperature. The nanocomposites containing nanoparticles were inactive at all mass ratios for the detection of methanol, and only one mass ratio (3:1:1) was able to detect ethanol, with a sensitivity of 4.3204 Ω ppm^−1^, and isopropanol, with a sensitivity of 3.4539 Ω ppm^−1^. These were also the highest sensitivities recorded using any of the sensors. All of the mass ratios for the ZnO nanorods were active for methanol and ethanol, but only one mass ratio (2:1:1) was able to detect isopropanol. The sensitivity values decreased as the mass ratio changed among 1:1:1, 2:1:1, and 3:1:1 when using ZnO NRs for the detection of methanol or ethanol. Regarding the sensors containing ZnO NFs, the 2:1:1 was inactive for the detection of methanol, while all mass ratios successfully detected ethanol, and only 1:1:1 could detect isopropanol. For the detection of methanol, only the 1:1:1 mass ratio with the ZnO NSs could detect the analyte. For ethanol and isopropanol, only the ZnO NS composite mass ratios of 1:1:1 and 2:1:1 detected the analytes. Also, the sensitivity decreased from the 1:1:1 to the 2:1:1 mass ratios. At a mass ratio of 1:1:1, only the ZnO NFs and NSs could detect all three analytes; at a mass ratio of 2:1:1, only the ZnO NSs can detect all three analytes.

In terms of the methanol vapour analyte, the shortest response time (1.88 min) and recovery time (1.86 min) were observed with the 3:1:1 ZnO NR composites. With ethanol, the shortest recovery time (1.20 min) was observed with the 3:1:1 ZnO NR composites. While the shortest response time (1.46 min) was observed with the 2:1:1 ZnO NF composite. The 3:1:1 ZnO NP composite has the shortest response time (1.73 min) with isopropanol, while the shortest recovery time (1.45 min) with isopropanol was observed with the 2:1:1 ZnO NS composites.

In general, morphologically anisotropic nanomaterials, specifically, ZnO NFs, and NSs show a better sensitivity for all three analytes, and ZnO NRs and NSs show a better performance in terms of their response and recovery time. Our future work will look at further refining anisotropic nanomaterials for optimal sensor performance.

## Figures and Tables

**Figure 1 nanomaterials-13-02810-f001:**
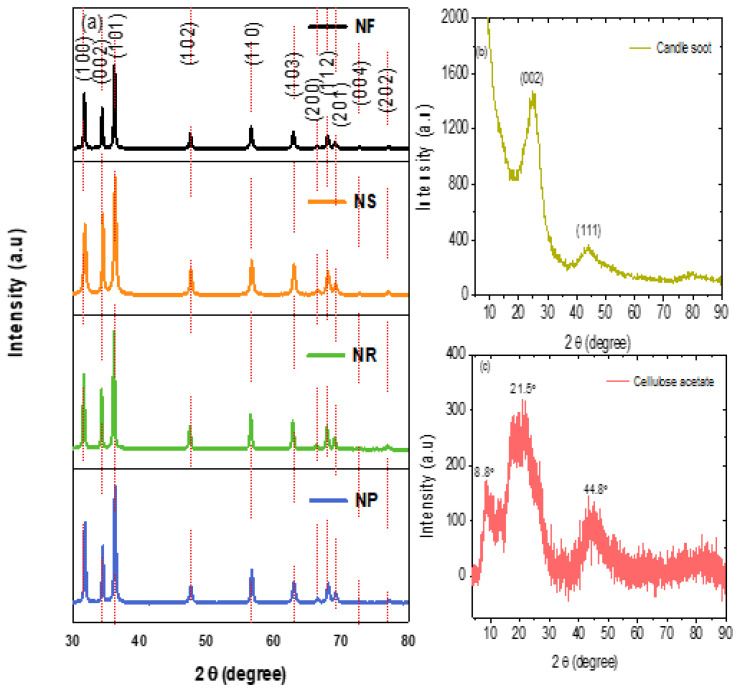
The XRD spectrum patterns of (**a**) the synthesised zinc oxide nanostructures (i.e., NF = nanoflower, NS = nanosheet, NR = nanorod, and NP = nanoparticle) (**b**) candle soot, and (**c**) purchased cellulose acetate, respectively.

**Figure 2 nanomaterials-13-02810-f002:**
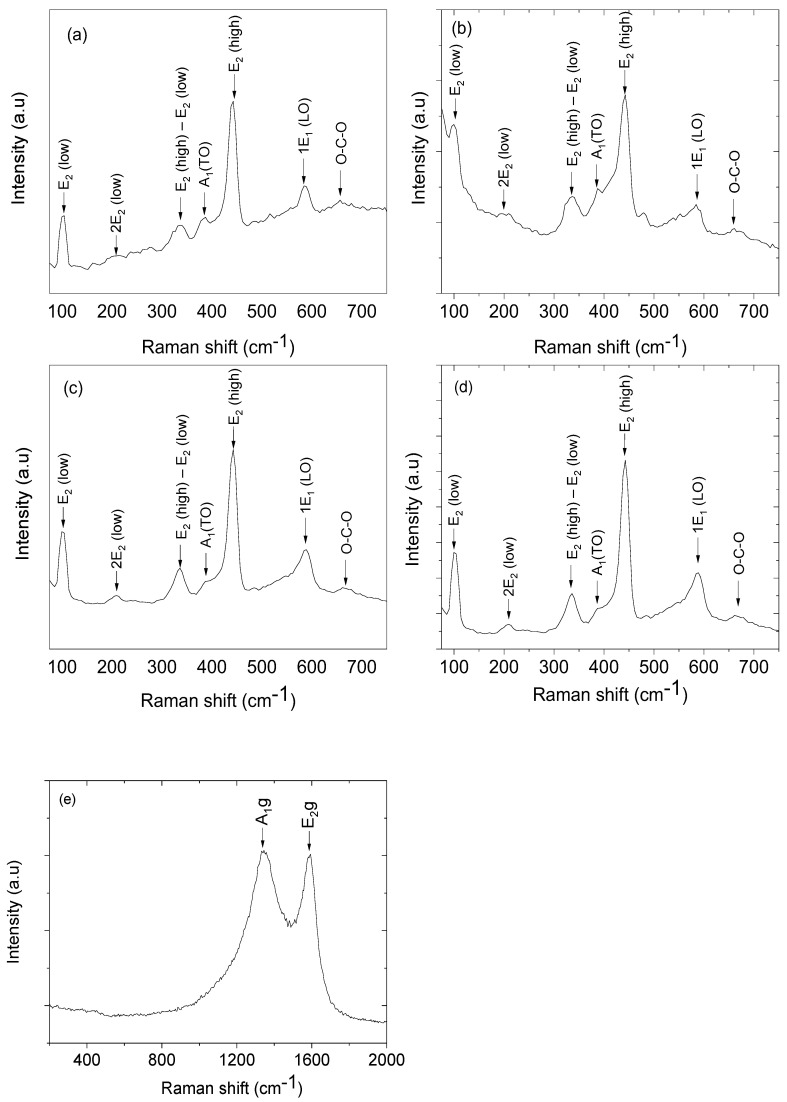
The Raman spectra of the synthesised ZnO nanostructures: (**a**) nanoflowers, (**b**) nanosheets, (**c**) nanorods, (**d**) nanoparticles, and (**e**) carbon nanoparticles.

**Figure 3 nanomaterials-13-02810-f003:**
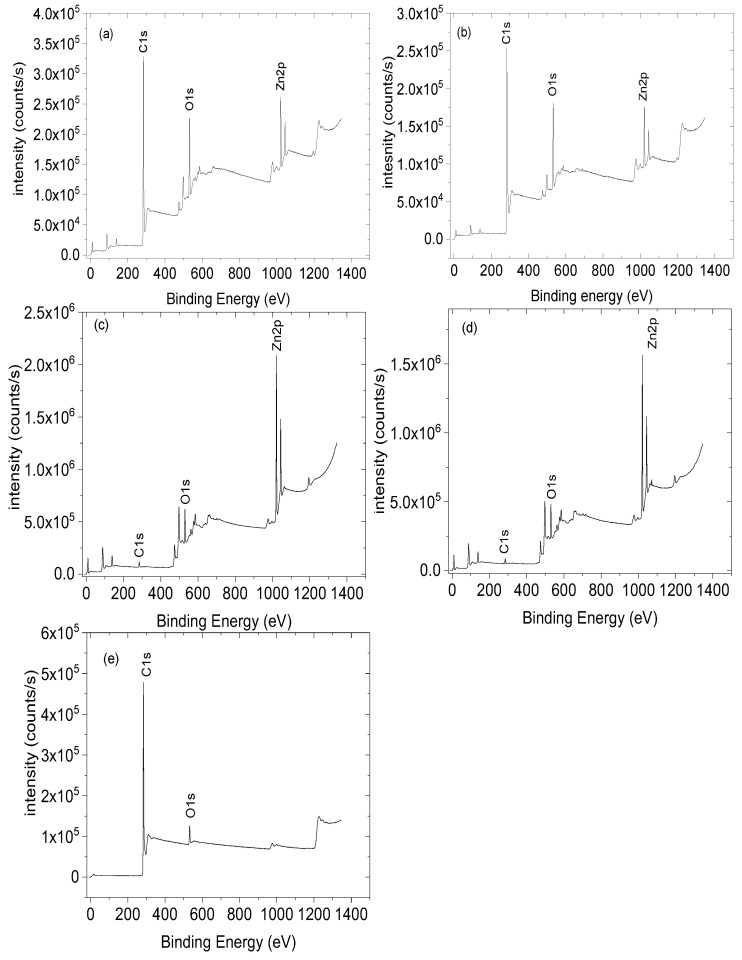
XPS spectra, XPS surveys of O 1s of (**a**) ZnO NRs, (**b**) ZnO NFs, (**c**) ZnO NSs, (**d**) ZnO NPs, (**e**) CNPs composites.

**Figure 4 nanomaterials-13-02810-f004:**
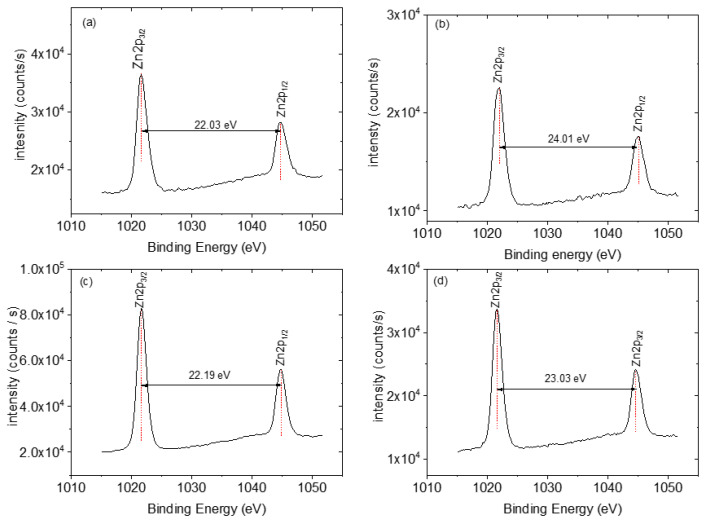
XPS spectra of (**a**) ZnO NPs, (**b**) ZnO NFs, (**c**) ZnO NSs, (**d**) ZnO NPs.

**Figure 5 nanomaterials-13-02810-f005:**
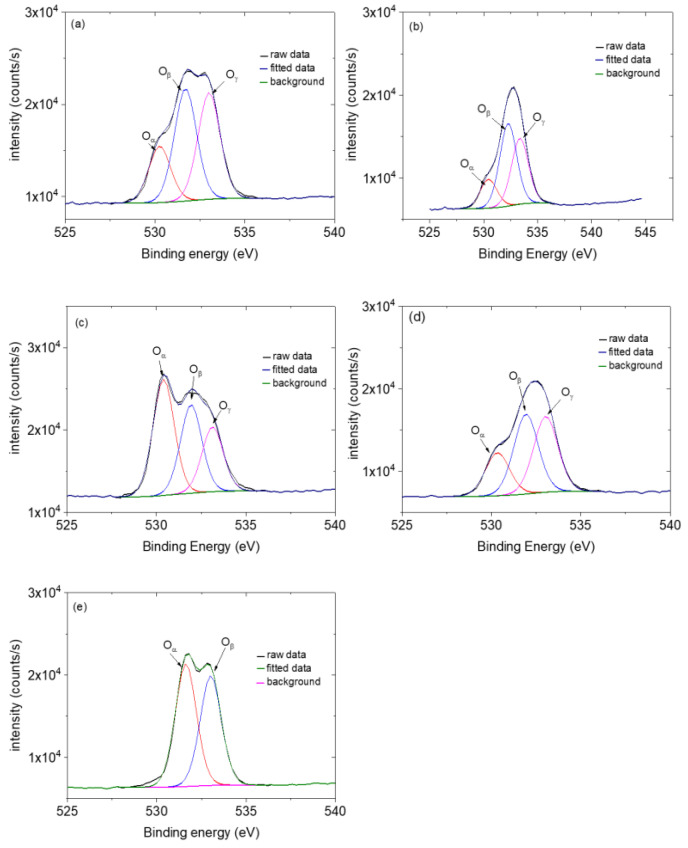
XPS survey of O 1s of (**a**) ZnO NRs, (**b**) ZnO NFs, (**c**) ZnO NSs, (**d**) ZnO NPs, (**e**) CNP composites.

**Figure 6 nanomaterials-13-02810-f006:**
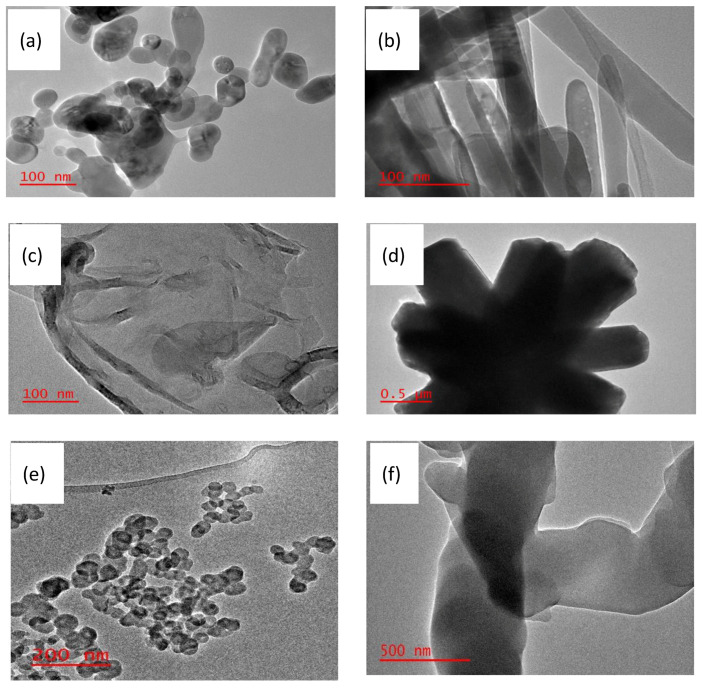
TEM images of: (**a**) ZnO nanoparticles (NPs), (**b**) ZnO nanorods (NRs), (**c**) ZnO nanosheets (NSs), (**d**) ZnO nanoflowers (NFs), (**e**) candle soot, and (**f**) purchased cellulose acetate, respectively.

**Figure 7 nanomaterials-13-02810-f007:**
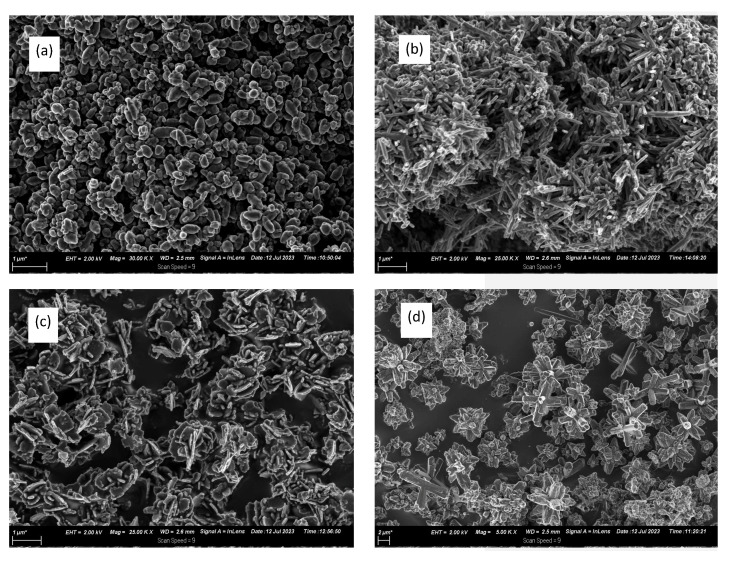
FE SEM images of (**a**) ZnO NPs, (**b**) ZnO NRs, (**c**) ZnO NSs, and (**d**) ZnO NFs.

**Figure 8 nanomaterials-13-02810-f008:**
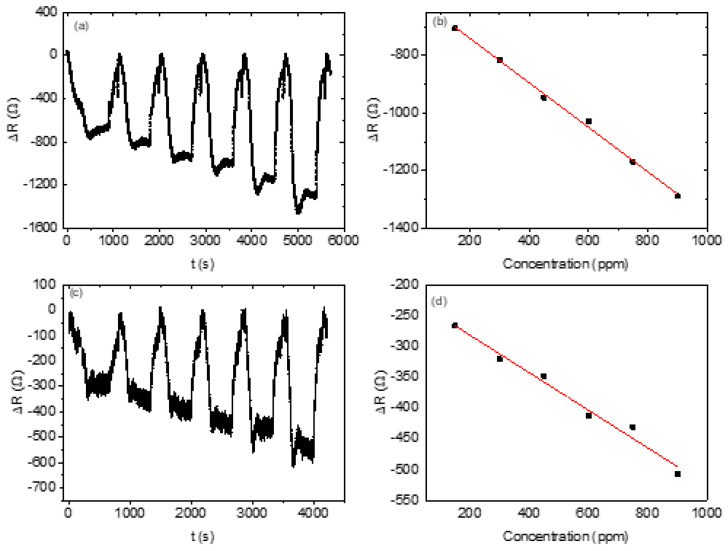
Dynamic response and recovery curves of methanol: (**a**,**b**) 1:1:1 mass ratio of ZnO nanorods; (**c**,**d**) 2:1:1 mass ratio of ZnO nanorods; (**e**,**f**) 1:1:1 mass ratio of ZnO nanosheets; (**g**,**h**) 1:1:1 mass ratio of ZnO nanoflowers; (**i**,**j**) 3:1:1 mass ratio of ZnO nanoflowers.

**Figure 9 nanomaterials-13-02810-f009:**
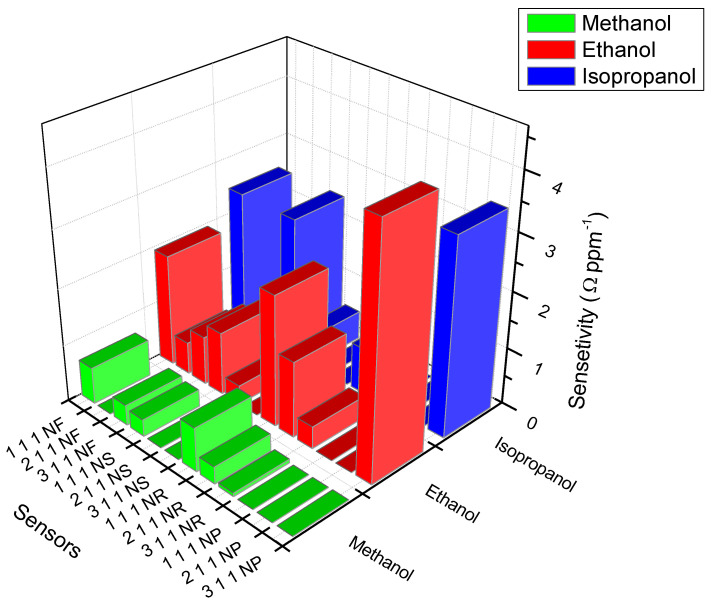
The sensitivity of the fabricated sensors towards methanol, ethanol, and is-propanol vapour.

**Table 1 nanomaterials-13-02810-t001:** A comparison of the Raman active modes of the various ZnO nanostructures with the literature.

Symmetry	Theoretical [44,45,46,49,50] (cm^−1^)	Nanoflowers (cm^−1^)	Nanosheets (cm^−1^)	Nanorods (cm^−1^)	Nanoparticles (cm^−1^)
$E_{2}^{low}$	101	102	100	103	100
*2* $E_{2}^{low}$	208	208	200	210	208
$E_{2}^{high}-E_{2}^{low}$	334	336	334	334	334
A_1_(TO)	380	384	390	389	387
E_1_(TO)	407	-	-	-	-
$E_{2}^{high}$	437	443	442	441	442
A1(LO)	574	-	-	-	-
1E_1_(LO)	583	585	586	587	587

**Table 2 nanomaterials-13-02810-t002:** Calculated atomic oxygen species percentages of the materials.

Nanomaterial	ZnO-NS	ZnO-NS-CNP-CA	ZnO-NP	ZnO-NP-CNP-CA	ZnO-NR	ZnO-NR-CNP-CA	ZnO-NF	ZnO-NF-CNP-CA	CNP
O_α_	64	31	65	19	54	26	56	12	59
O_β_	29	39	29	58	35	68	33	55	41
O_γ_	8	30	6	24	11	6	11	32	0

**Table 3 nanomaterials-13-02810-t003:** Textural characteristics of the synthesised ZnO nanostructure samples.

Sample Name	Surface Area m^2^/g	Pore Volume cm^3^/g	Pore Size Å
ZnO NPs	6.7	0.51787	251.06
ZnO NRs	12.7	0.22496	242.96
ZnO NSs	26.8	0.22284	202.87
ZnO NFs	1.3	0.19931	299.59
ZnO NP composite	32.0	0.38934	204.31
ZnO NR composite	32.9	0.29421	202.95
ZnO NS composite	36.7	0.35213	197.28
ZnO NF composite	30.1	0.23406	196.52

**Table 4 nanomaterials-13-02810-t004:** Summary of the performance of the fabricated sensors when detecting methanol vapour.

Sensor Name	Sensitivity (Ω ppm^−1^)	Response Time (Min.)	Recovery Time (Min.)
Nanoparticles (3:1:1)	0	0	0
Nanorods (3:1:1)	0.06437	1.88	1.86
Nanosheets (3:1:1)	0	0	0
Nanoflowers (3:1:1)	0.3572	2.43	2.95
Nanoparticles (2:1:1)	0	0	0
Nanorods (2:1:1)	0.3048	2.48	2.23
Nanosheets (2:1:1)	0	0	0
Nanoflowers (2:1:1)	0	0	0
Nanoparticles (1:1:1)	0	0	0
Nanorods (1:1:1)	0.7740	4.20	2.95
Nanosheets (1:1:1)	0.2994	2.00	3.60
Nanoflowers (1:1:1)	0.6485	2.60	3.54

## Data Availability

Data is available when request made.

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
