# Peer review of "Exploration of the Different Dimensions of Wurtzite ZnO Structure Nanomaterials as Gas Sensors at Room Temperature"

_nanomaterials, 2023, doi:10.3390/nano13202810_

Round 1
Reviewer 1 Report
This paper presents a study on ZnO-based gas sensors for detecting primary alcohols. The study primarily examines how the sensor's response is affected by the morphologies of ZnO. The dependence of device responses on the morphology of ZnO has been widely studied in various fields of materials science, including sensors. This study is particularly interesting because it demonstrates the ability to detect primary alcohols at room temperature and atmospheric pressure. However, this manuscript marginally meets the requirements for publication in Nanomaterials due to the following major concerns.
1. The authors synthesize ZnO nanostructures following known procedures and hybridize them with carbon nanoparticles obtained from candle soot. The obtained nanostructures have been analyzed using different techniques. However, most of the hybrid/composite materials have hardly been characterized and verified. It would have been of great interest to see the characterization of all the hybrid materials. Mainly since the focus of this work is on the hybrids of different morphologies of ZnO nanostructures with carbon nanoparticles, the manuscript without the characterizations of which looks too raw.
2. A crucial component missing in the manuscript is the BET isotherm, which makes it hard to verify the data in Table 4.
The language requires editing and a few scientific terms need to be corrected. For example, on page 1, line 27, "electron-denoting" should be changed to "electron-donating".
Author Response
Dear Reviewers,
We appreciate the time and energy you spent reviewing our manuscript; we found your comments and suggestions very useful to improve our manuscript, and we considered all the suggestions and comments and incorporated them into the revised manuscript. All the changes are in RED in the main text.
Review 1
Open Review
Comments and Suggestions for Authors
This paper presents a study on ZnO-based gas sensors for detecting primary alcohols. The study primarily examines how the sensor's response is affected by the morphologies of ZnO. The dependence of device responses on the morphology of ZnO has been widely studied in various fields of materials science, including sensors. This study is particularly interesting because it demonstrates the ability to detect primary alcohols at room temperature and atmospheric pressure. However, this manuscript marginally meets the requirements for publication in Nanomaterials due to the following major concerns.
- The authors synthesize ZnO nanostructures following known procedures and hybridize them with carbon nanoparticles obtained from candle soot. The obtained nanostructures have been analyzed using different techniques. However, most of the hybrid/composite materials have hardly been characterized and verified. It would have been of great interest to see the characterization of all the hybrid materials. Mainly since the focus of this work is on the hybrids of different morphologies of ZnO nanostructures with carbon nanoparticles, the manuscript without the characterizations of which looks too raw.
Authors response: the hybrid materials were synthesized by physical mixing, and from our previous studies, we didn't observe new diffraction patterns for the case of XRD; no new IR bands/shifts other than the individual materials patterns and IR shifts. However, for sensor application, the reactive oxygen on the hybrid materials and the pore size of the individual and the hybrid materials are very important, so we included the XPS and BET results in the manuscript.
- A crucial component missing in the manuscript is the BET isotherm, which makes it hard to verify the data in Table 4.
Authors response: the BET isotherm was included in the supplementary documents, Figure S3
Comments on the Quality of English Language
The language requires editing and a few scientific terms need to be corrected. For example, on page 1, line 27, "electron-denoting" should be changed to "electron-donating".
Author response: the typo error was corrected
Reviewer 2 Report
The article "Exploration of the different dimensions of the wurtzite ZnO structure nanomaterial as gas sensors at room temperature" describes the synthesis of ZnO with various morphologies, used to obtain composite sensors for organic compounds. It is a valuable study that can be published after authors address the following problems:
Abstract should be checked and revised carefully by briefly introducing the work plan and key findings.
Abstracts should highlight the innovation of the article, as often abstract section is presented separately in search engines, it must be able to stand alone as an informative piece. In the abstract, need to focus more on the quantitative information, not qualitative one (rows 14-17 could benefit from actual information and values rather than generic description).
The English language needs some polishing for style and typos (e.g. use proper indices in section 3.2 for cm-1 and section 3.6 for ppm-1 and row 536 Zn(CH3COO)2·2H2O; row 396 “are responded”; row 424 use nano with small letter like in the case of other nanostructures; authors used both nano flower and nanoflower;
This work is interesting and can be boosted further. I suggest that the authors find an opportunity to mention the following publications, could prove this manuscript, in introduction doi: 10.3390/pharmaceutics14122842; doi: 10.3390/ijms24065677; doi: 10.1016/j.ceramint.2022.11.178
In the XRD section the peak from 76.6o is not corresponding to (200) but to (202) Miller indices. Please correct the description in text and Figure 1. To improve the readability of Figure 1a, I suggest to present the XRD for ZnO between 30 - 80o as there are no characteristic peaks outside this interval, and use other colour than yellow (maybe red) for the NF sample.
In FTIR image (Figure S2a) please shift the left marker for COOH moiety so that will be over the peaks. Please check the above recommended articles for the full assignment of minor peaks.
In Figure 2 please use uniform size of subfigures and font. At this moment some have intensity other Intensity, and Raman is either bold or not, with different size.
The conclusion should reflect the heuristic of the study. How is this system a better one? Conclusion section must be reworked to underline the novelty and advantages of this research, with actual numbers.
Author Response
Dear Reviewers,
We appreciate the time and energy you spent reviewing our manuscript; we found your comments and suggestions very useful to improve our manuscript, and we considered all the suggestions and comments and incorporated them into the revised manuscript. All the changes are in RED in the main text.
Review 2
Comments and Suggestions for Authors
The article "Exploration of the different dimensions of the wurtzite ZnO structure nanomaterial as gas sensors at room temperature" describes the synthesis of ZnO with various morphologies, used to obtain composite sensors for organic compounds. It is a valuable study that can be published after authors address the following problems:
Abstract should be checked and revised carefully by briefly introducing the work plan and key findings.
Abstracts should highlight the innovation of the article, as often abstract section is presented separately in search engines, it must be able to stand alone as an informative piece. In the abstract, need to focus more on the quantitative information, not qualitative one (rows 14-17 could benefit from actual information and values rather than generic description).
Authors' response: the abstract was edited thoroughly, and more information about the key findings, including the numerical values, was included (please see the Abstract section).
The English language needs some polishing for style and typos (e.g. use proper indices in section 3.2 for cm-1 and section 3.6 for ppm-1 and row 536 Zn(CH3COO)2·2H2O; row 396 "are responded"; row 424 use nano with small letter like in the case of other nanostructures; authors used both nano flower and nanoflower;
Authors' response: all typos and errors were corrected and highlighted in red in the main text.
This work is interesting and can be boosted further. I suggest that the authors find an opportunity to mention the following publications, could prove this manuscript, in introduction doi: 10.3390/pharmaceutics14122842; doi: 10.3390/ijms24065677; doi: 10.1016/j.ceramint.2022.11.178
Authors' response: thank you; we included the most relevant one in the reference section.
In the XRD section the peak from 76.6o is not corresponding to (200) but to (202) Miller indices. Please correct the description in text and Figure 1. To improve the readability of Figure 1a, I suggest to present the XRD for ZnO between 30 - 80o as there are no characteristic peaks outside this interval, and use other colour than yellow (maybe red) for the NF sample.
Authors' response: thank you very much, the correction was included in the main text. Figure 1 has improved; the x-axis scale was adjusted to 30-80o the color of the NF is changed to black.
In FTIR image (Figure S2a) please shift the left marker for COOH moiety so that will be over the peaks. Please check the above recommended articles for the full assignment of minor peaks.
Authors' response: the graphs were redrawn, and all major peaks were indicated correctly.
In Figure 2 please use uniform size of subfigures and font. At this moment some have Intensity other Intensity, and Raman is either bold or not, with different sizes.
Authors' response: thank you very much. We have edited the figure to have a uniform font size.
The conclusion should reflect the heuristic of the study. How is this system a better one? Conclusion section must be reworked to underline the novelty and advantages of this research, with actual numbers.
Authors' response: the conclusion section was rewritten; the study's novelty, key findings, and the study's advantages with numerical values were added. (please see the conclusion section)
Reviewer 3 Report
This presented study focused on semiconductor metal oxide-based gas sensors, widely used for detecting toxic gases. Authors synthesized four distinct zinc oxide (ZnO) morphologies (nanoparticles, nanorods, nanosheets, and nanoflowers) from a single precursor, Zn(CH3COO)2·2H2O, by modifying reaction conditions. Physicochemical properties assessment, including surface area measurement, and pore size analysis. The materials were integrated with carbon nanoparticles and cellulose acetate (CA) to form composites, which acted as sensing materials in solid-state sensors to detect MeOH, EtOH, and isopropanol vapors at room temperature. Sensor responses, measured as relative resistance, varied based on material morphology, composite mass ratio, and analyte type. Overall, the presented work is interesting, however requires substantial revision before accepted for publication,
Major concerns,
1- One of the important aspects, ignored by the authors is the interaction of MeOH or EtOH to ZnO NPs surface at room temperature. Such interactions at room temperature can form irreversible MeOH (such as methoxy) or EtOH species over the ZnO NPs (sensing material), changing the sensor baseline, as obvious in Figure 8 and supplementary Figures S4 to S6. Such baseline shifts can permanently alter the sensing performance of ZnO NPs, which needs to be explained. A quick literature scan shows that such interactions, changing the sensors baseline are well studied, such as doi.org/10.1039/D2CY01395A. Authors need to discuss their results (baseline shift in Figure 8 and S4-S6), in the light of the suggested article.
2- Additionally, all the sensor reproducibility should be tested, both at low and high concentrations (ppm) of MeOH, EtOH, and isopropanol.
Minor comments,
1- The abstract lacks the main findings of the study, and needs revision, Please use obtained values (statistical) to summarize the key findings.
2- Please clarify the novelty aspect of the study in the abstract and introduction last paragraph.
3- Figure 1a, please do the structural refinement of the XRD data.
4- Figure S2, it is not possible to precise the FTIR bands (ZnO) at <1000 cm-1. Please also explain how the smoothing of the data and baseline correction was performed.
5- Figure 5, what is the goodness of peak fitting score (R2) and adopted method?
6- Figure 7 (c to f), the morphologies of the ZnO NPs are not clear. The claim NPs-morphologies should be clarified. Please add alternative TEM images or SAED pattern.
7- Figure 8, scale bar is not readable. Please revise.
8- Figure 9, axis is not readable. Please revise.
9- Conclusion part is similar to the abstract, please add future perspective.
In conclusion, I recommend Major revision.
Author Response
Dear Reviewers,
We appreciate the time and energy you spent reviewing our manuscript; we found your comments and suggestions very useful to improve our manuscript, and we considered all the suggestions and comments and incorporated them into the revised manuscript. All the changes are in RED in the main text.
Review 3
Comments and Suggestions for Authors
This presented study focused on semiconductor metal oxide-based gas sensors, widely used for detecting toxic gases. Authors synthesized four distinct zinc oxide (ZnO) morphologies (nanoparticles, nanorods, nanosheets, and nanoflowers) from a single precursor, Zn(CH3COO)2·2H2O, by modifying reaction conditions. Physicochemical properties assessment, including surface area measurement, and pore size analysis. The materials were integrated with carbon nanoparticles and cellulose acetate (CA) to form composites, which acted as sensing materials in solid-state sensors to detect MeOH, EtOH, and isopropanol vapors at room temperature. Sensor responses, measured as relative resistance, varied based on material morphology, composite mass ratio, and analyte type. Overall, the presented work is interesting, however requires substantial revision before accepted for publication,
Major concerns,
1- One of the important aspects, ignored by the authors is the interaction of MeOH or EtOH to ZnO NPs surface at room temperature. Such interactions at room temperature can form irreversible MeOH (such as methoxy) or EtOH species over the ZnO NPs (sensing material), changing the sensor baseline, as obvious in Figure 8 and supplementary Figures S4 to S6. Such baseline shifts can permanently alter the sensing performance of ZnO NPs, which needs to be explained. A quick literature scan shows that such interactions, changing the sensors baseline are well studied, such as doi.org/10.1039/D2CY01395A. Authors need to discuss their results (baseline shift in Figure 8 and S4-S6), in the light of the suggested article.
Authors response: thank you very much. The sensor response is facing down (the resistance decreases as the concentration increases). The sensors' responses are baseline-corrected (it is in ΔR). The cause of drift in sensor response has many factors; however, the physicochemical interaction between the sensing materials and the analytes is the most dominant besides the measurement instruments' stability. As you indicated, the analyte molecules adsorbed reversibly on the surface of the sensing materials cause the drift; in some instances, permanent drifting occurs. In our previous studies, we indicated the adsorption of the analyte molecules is reversible, DOI: 10.1016/j.rinp.2023.106864, DOI: 10.1039/D3NA00050H, doi.org/10.1063/5.0063604, 10.1007/s10854-018-00633-x
2- Additionally, all the sensor reproducibility should be tested, both at low and high concentrations (ppm) of MeOH, EtOH, and isopropanol.
Authors response: the result of reproducibility of one of the sensors was included in the additional information section as an example as Figure S8.
Minor comments,
1- The abstract lacks the main findings of the study and needs revision,Please use obtained values (statistical) to summarize the key findings.
Authors response: The abstract has been reworked as suggested. The abstract was edited thoroughly, and more information about the key findings, including the numerical values, was included (please see the Abstract section).
2- Please clarify the novelty aspect of the study in the abstract and introduction last paragraph.
Authors response: The abstract has been reworked and additionally noted the sensors were operated at room temperature without an additional heat source
3- Figure 1a, please do the structural refinement of the XRD data.
Authors response: The XRD data was processed using the in-buitl software on the instrument
4- Figure S2, it is not possible to precise the FTIR bands (ZnO) at <1000 cm-1. Please also explain how the smoothing of the data and baseline correction was performed.
The data was baseline corrected and normalized. The following sentence was added in L154 “The obtained data was baseline-corrected and normalized.”
5- Figure 5, what is the goodness of peak fitting score (R2) and adopted method?
Authors' response: the curve fittings were identified as raw data, fitting total and background with different colours. The Gaussian function was used to fit the curves (indicated under the section of XPS)
6- Figure 7 (c to f), the morphologies of the ZnO NPs are not clear. The claim NPs-morphologies should be clarified. Please add alternative TEM images or SAED pattern.
Authors' response: thank you for the comment; we have inserted better images for the ZnO morphologies; for the NF ZnO Fig 6 (d) it is 3D, and it isn't easy to have clearer images from the TEM instruments, hoverer the FE SEM can support the result much better than the TEM images.
7- Figure 8, scale bar is not readable. Please revise.
Authors' response: the image is magnified, and now it is visible.
8- Figure 9, axis is not readable. Please revise.
Authors' response: the chart was redrawn and the axises are visible
9- Conclusion part is similar to the abstract, please add future perspective.
Authors' response: the conclusion section was rewritten; the study's novelty, key findings, and the study's advantages with numerical values were added. (please see the conclusion section)
In conclusion, I recommend Major revision.
Round 2
Reviewer 1 Report
The authors have significantly improved the scientific content of the manuscript, making it suitable for publication in Nanomaterials.
Decision: Accept
Reviewer 2 Report
The authors thoroughly addressed my comments in the 1st round of the revision. I think this manuscript is suitable for publication in Nanomaterials.
Reviewer 3 Report
I can see that the manuscript is improved after revision, so can be accepted for publication.